# A Label-Free Proteomic Approach for the Identification of Biomarkers in the Exosome of Endometrial Cancer Serum

**DOI:** 10.3390/cancers14246262

**Published:** 2022-12-19

**Authors:** Eduardo Sommella, Valeria Capaci, Michelangelo Aloisio, Emanuela Salviati, Pietro Campiglia, Giuseppe Molinario, Danilo Licastro, Giovanni Di Lorenzo, Federico Romano, Giuseppe Ricci, Lorenzo Monasta, Blendi Ura

**Affiliations:** 1Department of Pharmacy, University of Salerno, 84084 Salerno, Italy; 2Institute for Maternal and Child Health, IRCCS Burlo Garofolo, 34137 Trieste, Italy; 3Life Sciences Department, University of Trieste, 34149 Trieste, Italy; 4AREA Science Park, Basovizza, 34149 Trieste, Italy; 5Department of Medical, Surgical and Health Sciences, University of Trieste, 34149 Trieste, Italy

**Keywords:** exosome, endometrial cancer, biomarker, LFQ-MS

## Abstract

**Simple Summary:**

Endometrial cancers (ECs) are mostly adenocarcinomas arising from the inner part of the uterus. The identification of serum biomarkers may be useful for an early diagnosis. This study compared the exosome serum proteins of 12 patients with EC with those of 12 non-cancer subjects, and identified 33 proteins with diagnostic potential. Quantification analysis in 36 patients with endometrial cancer compared to 36 healthy individuals confirmed the upregulation of APOA1, HBB, CA1, HBD, LPA, SAA4, PF4V1, and APOE. We developed a statistical model based on this set of proteins that detects cancer samples with excellent sensitivity and specificity levels, particularly for stage 1 ECs. In our opinion, the combined levels of PF4V1, CA1, HBD, and APOE have great potential to reach the clinical stage, after a validation phase.

**Abstract:**

Endometrial cancers (ECs) are mostly adenocarcinomas arising from the inner part of the uterus. The identification of serum biomarkers, either soluble or carried in the exosome, may be useful in making an early diagnosis. We used label-free quantification mass spectrometry (LFQ-MS)-based proteomics to investigate the proteome of exosomes in the albumin-depleted serum from 12 patients with EC, as compared to 12 healthy controls. After quantification and statistical analysis, we found significant changes in the abundance (*p* < 0.05) of 33 proteins in EC vs. control samples, with a fold change of ≥1.5 or ≤0.6. Validation using Western blotting analysis in 36 patients with EC as compared to 36 healthy individuals confirmed the upregulation of APOA1, HBB, CA1, HBD, LPA, SAA4, PF4V1, and APOE. A multivariate logistic regression model based on the abundance of these proteins was able to separate the controls from the EC patients with excellent sensitivity levels, particularly for stage 1 ECs. The results show that using LFQ-MS to explore the specific proteome of serum exosomes allows for the identification of biomarkers in EC. These observations suggest that PF4V1, CA1, HBD, and APOE represent biomarkers that are able to reach the clinical stage, after a validation phase.

## 1. Introduction

Endometrial cancers (ECs) are tumors of the inner part of the uterus, deriving mostly from the glandular tissue. ECs account for 75–80% of all uterine cancers, and represent one of the most common gynecologic malignancies, affecting ∼3% of women [1]. The risk of EC development is associated with genetic predisposition (e.g., Lynch syndrome), racial background, age, obesity, metabolic syndrome, diabetes, polycystic ovary syndrome, and high estrogen exposure. Metabolic conditions related to lifestyle represent the most significant risk factors [2,3,4,5,6,7]. Currently, EC incidence and mortality are continuously rising, with the highest rates registered in North America and Europe [8].

Based on epidemiological, clinical, pathological, and molecular characteristics, ECs are classified into two different subtypes. Type I ECs represent the majority of new cases, usually low-grade and associated with a favorable prognosis. Common features of type I ECs are endometrioid cell morphology resembling the normal endometrium (low-grade, I or II), and expression of estrogen receptors [9,10]. Type II ECs are more aggressive forms, usually hormone-receptor-negative high-grade tumors with poor prognoses. Histologically, type II ECs are mainly characterized by being serous and clear cell undifferentiated carcinomas and carcinosarcomas [9,10].

At clinical presentation, EC symptoms are poorly specific, which include abnormal uterine bleeding as the most common symptom in advanced stages in post-menopause women. In advanced EC, bleeding is often accompanied by abdominal and pelvic pain [11]. Currently, there are no specific tests for the diagnosis of EC; standard diagnostic protocols consist of pelvic ultrasonography, and a certain diagnosis commonly requires invasive investigation procedures, such as hysteroscopy, with targeted biopsy or blinded dilation and curettage (D & C) [12]. Expanding biological understanding of EC and refining its molecular characterization is still challenging, but such strategies may uncover new exploitable biomarkers. The use of proteomic techniques is making a great contribution to understanding the pathophysiology of tumors [13]. Using proteomic methodologies allows us to identify putative biomarkers in different cancers, including gastric cancer [14], liver cancer [15], and ovarian cancer [16]. In EC, numerous biomarkers have been reported from different biological samples (including tumor tissue, blood, serum, plasma, urine, and uterine lavage fluid), but none have reached the clinical stage thus far ([17,18], Njoku 2019). The identification of circulating biomarkers may allow for easy detection, avoiding invasive sampling procedures. Thanks to a proteomic study, we recently identified new putative biomarkers in the sera of EC patients (namely SBSN [19], CLU, SERPINC1, ITIH4, C1R [20], GAL-9, GAL-1, MMP7, FASLG, and COL9A1 [21]).

A growing body of evidence has shown that tumor cells release a plethora of extracellular vesicles (EVs) that carry lipids, proteins, DNA, mRNAs, and miRNAs [22]. Among them, the best characterized are exosomes, microvesicles that originate in the multivesicular body and measure 30–100 nm in diameter; they are released in body fluids, including blood, bile, urine, and saliva [23]. From a clinical perspective, finding key mediators of EC released in exosomes may allow us to identify potential biomarkers of cancer development and progression. By applying LC-MS/MS, Song Y and colleagues recently identified LGALS3BP as a possible biomarker in the plasmatic exosome of EC [24]. Here, using a label-free quantification (LFQ) proteomics approach, the present study aimed to characterize the exosomal proteome from the albumin-depleted serum, in order to identify novel EC biomarkers.

## 2. Materials and Methods

### 2.1. Patients

For proteomic and data validation, a total of 72 patients (36 women suffering from EC, and 36 non-EC controls) were recruited at the Institute for Maternal and Child Health—IRCCS “Burlo Garofolo” (Trieste, Italy) during 2019 and 2021. All procedures complied with the Declaration of Helsinki, and were approved by the Institutional Review Board of IRCCS Burlo Garofolo and the Regional Ethics Committee (CEUR-2020-Os-030). All patients signed informed consent forms. The clinical and pathological characteristics of the patients are described in Appendix A Appendix A. The median age of the patients was 68 years (Interquartile Range: IQR 57–75; Min = 48, Max = 88), while the median age of the controls was 38 (IQR 28–54; Min = 23, Max = 78). Subjects who were positive to human immunodeficiency virus (HIV) or hepatitis B or C virus (HBV, HBC) were excluded from the study. The controls who were chosen excluded oncologic patients, and patients with leiomyomas or adenomyosis. In this study, we also excluded controls with benign tumors (myoma), chronic inflammatory disease (adenomyosis), or viral infections, since these pathologies may affect the abundance of proteins in serum exosome analysis.

### 2.2. Serum Sample Collection and EV Isolation

Serum was obtained with blood centrifugation at 5000 rcf for 5 min, and was collected and stored at −80 °C. In order to improve the proteomic study, 100 µL of crude serum was incubated for 5 min with the Albumin Depletion Kit (Thermo Fisher, Waltham, MA, USA). After column elution, EVs were isolated with the Total Exosome Isolation kit (Thermo Fisher Scientific), as reported in [20]. Then, 100 µL of depleted serum was mixed with 20 µL reagent and incubated for 30 min at 4 °C. After incubation, the samples were centrifuged at 10,000 rcf for 10 min at room temperature, and resuspended with 30 µL of PBS. Sample characterization was performed, as previously reported in [18]. Protein content was determined using Bradford reagent.

### 2.3. Exosome Digestion and MS Analysis

An amount of 100 µg of depleted exosomes was digested with the EasyPep™ MS Sample Prep Kits (Thermo Fisher). After digestion, analysis was performed with nanoflow ultra-high performance liquid chromatography high-resolution mass spectrometry, using an Ultimate 3000 nanoLC (Thermo Fisher Scientific, Bremen, Germany) coupled to an Orbitrap Lumos tribrid mass spectrometer (Thermo Fisher Scientific) that used a nanoelectrospray ion source (Thermo Fisher Scientific). A volume of 1 μL of digestion was initially trapped on a PepMap trap column for 1.50 min at a flow rate of 30 μL/min (Thermo Fisher), and then peptides were loaded and separated onto a C18-reversed phase column (250 mm × 75 μm I.D, 2.6 µm, 100 Å, BioZen Phenomenex, Bologna, Italy). The flow rate was set to 300 nL/min. The obile phases were A): 0.1% HCOOH in water *v*/*v*, and B): 0.1% HCOOH in ACN/Water *v*/*v* 80/20. A linear 60 min gradient was performed. The samples were run in duplicate. HRMS analysis was performed in data-dependent acquisition (DDA), with an MS1 range of 375–1500 *m*/*z*; HCD fragmentation was used with normalized collision energy setting 27. Resolution was set at 120,000 for MS1 and 15,000 for MS/MS. Single and unassigned charges were excluded. Quadrupole isolation was set to 3Da. The maximum ion injection times for MS (OT) and the MS/MS (OT) scans were set to auto and to 60 ms, respectively, and ACG values were set to standard. The dynamic exclusion was 30 s. For data processing, raw MS data were analyzed using the Mascot Distiller 2.8 with the Mascot search engine. The MS/MS scans were matched against the human proteome (Uniprot 03/2022 version). The following parameters were used: enzyme trypsin, missed cleavages max 1, mass accuracy tolerance 10 ppm and 0.6 Da for precursors and fragments, respectively. Carbamidomethylcysteine was used as fixed modification, while methionine oxidation as variable. Proteins were considered identified with at least one unique peptide setting a false discovery threshold of <1%. The label-free quantification was performed with Mascot Distiller software, based on the Replicate protocol. This workflow was based on the relative intensities of high-resolution extracted ion chromatograms (XICs) for precursor ions in multiple data sets, and aligned using mass and elution time. Relative quantitation was based on protein ratio calculation, which uses the median of the assigned peptide ratios. The minimum precursor charge was set to 2, and the minimum peptides number was set to 2. The Replicate protocol was used to measure the relative abundance of a protein from sample to sample.

### 2.4. Western Blotting

Western blotting in serum exosomes was performed, as previously described [25]. For this analysis, 30 µg of proteins were loaded on precast gel 4–20%, and then transferred to a nitrocellulose membrane; then, the membrane was blocked with 5% defatted milk in TBS-Tween 20 and incubated overnight at 4 °C, with antibodies against CD9 (1:800, rabbit polyclonal), CD64 (1:1000, rabbit polyclonal), LPA (1:800, rabbit polyclonal), PF4V1 (1:2000, rabbit polyclonal), HBD (1:1200, rabbit polyclonal), SAA4 (1:300, rabbit polyclonal), HBB (1:300 mouse monoclonal), CA1 (1:800, rabbit polyclonal), APOE (1:800, rabbit polyclonal), and APOA1 (1:800, rabbit polyclonal). After incubation with primary antibodies, the membranes were incubated with secondary antibodies HRP-conjugated anti-rabbit IgG or anti-mouse IgG (1:3000, Sigma-Aldrich, St. Louis, MO, USA; Merck KGaA, Darmstadt, Germany). The presence of the same proteins was confirmed in the cancer tissue. Western blotting was performed in 30 µg of total lysates from 3 tumor tissues. Tissues were lysed in 1% NP-40, 50 mM Tris-HCl (pH 8.0), NaCl 150 mM with Phosphatase Inhibitor Cocktail Set II 1× (Millipore, Burlington, VT, USA) and 2 mM phenylmethylsulfonyl fluoride (PMSF), and 1 mM benzamidine.

SuperSignal West Pico Chemiluminescent was used for protein band signal visualization. The intensity of the immunostained bands was quantified, normalizing on the total protein content evaluated by the red ponceau solution staining of the membrane from the same blot.

### 2.5. Bioinformatic Analysis

Proteins identified through MS were analyzed with gProfiler classification systems, and categorized according to their molecular function involvement, biological processes, and protein class. Pathway analysis was carried out using the Reactome tool. The bio-functions were generated via Ingenuity Pathway Analysis (IPA) [26]. Results from the IPA were considered statistically significant when *p* < 0.01. For the filter summary, we only considered associations where the confidence was high (predicted), or those that had been observed experimentally.

### 2.6. Statistical Analysis

Differences were considered significant between patients and controls when proteins showed a fold change of ±1.5, and satisfied the Mann–Whitney rank sum test (*p* < 0.05). All analyses were conducted with Stata/IC 16.1 for Windows (StataCorp LP, College Station, TX, USA). Proteins with a fold change ≥ 3.5 and *p* < 0.05 were further validated. With this selected number of proteins, we elaborated a predictive model using a multivariate logistic regression approach. The dependent variable was EC patients vs. controls. Independent variables were the exosomes selected as explained above. We adopted a step-down procedure, and thus began with all proteins and excluded, one at a time, those which had the highest *p*-values if *p* ≥ 0.05. Consequently, the resulting model comprised only exosomal proteins that were significantly associated with the outcome. For this final model, we reported the coefficients of the predictive probability function, the Area under the Receiver Operating Characteristic (ROC) curve (AUC), and levels of sensitivity and specificity. Finally, given that the results of the EC patients vs. controls model were not fully satisfactory, we elaborated two different models using the same multivariate logistic regression, but considered stage 1 EC patients vs. controls and advanced stage (2, 3, and 4) EC patients vs. controls separately.

## 3. Results

### 3.1. Proteomic Analysis Reveals a Specific Exosomal Proteome in EC Patients

To characterize the exosomal proteome of EC patients’ sera, we first isolated exosomes from the albumin-depleted sera of 12 ECs in stage 1 and 12 controls. To verify proper exosome isolation, we performed Western blotting (Appendix A) of common exosome markers CD 63 and CD9 (Figure 1).

Next, to investigate the whole proteomic profile of these ECs as compared to the controls, we subjected the obtained exosomes to LFQ nanoLC–MS/MS-based proteomic analysis. We identified 421 exosomal protein groups, with one unique peptide and FDR < 1%. After quantification and statistical analysis, we found that 33 proteins showed a significant alteration (*p* < 0.05) in their abundance in EC vs. control samples, (Table 1). In detail, 31 proteins displayed increased levels in EC samples (fold change ≥ 1.5), while only two showed lower levels (fold change ≤ 0.6). The accession numbers, gene names, score, and peptide numbers are listed in Appendix A Appendix A.

### 3.2. Western Blotting Validation of Eight Exosomal Proteins in EC Patients

To validate data obtained from the proteomic analysis, we focused on those proteins that showed the strongest differences in abundance between the two groups. Thus, starting from the 33 identified proteins, we selected eight (namely APOA, HBB, CA1, HBD, LPA, SAA4, PF4V1, and APOE) that showed a fold change ≥ 3.5 and a *p*-value < 0.05. The abundance of these proteins was then validated with Western blotting in 36 ECs versus 36 controls. For the validation, we analyzed exosomes that derived from EC patients at different stages, with the first 18 patients at stage 1 and the remaining 18 at advanced stages.

Western blotting (Appendix A) quantitative analysis confirmed the LC-MS/MS results, showing a higher serum abundance in EC patients than in controls, for all proteins. The higher abundance was significant for APOA1 (*p* = 0.04) (Figure 2A), HBB (*p* = 0.0073) (Figure 2B), CA1 (*p* = 0.0038) (Figure 2C), HBD (*p* = 0.015) (Figure 2D), LPA (*p* = 0.02) (Figure 2E), SAA4 (*p* = 0.028) (Figure 2F), and PF4V1 (*p* = 0.0093) (Figure 2G), while APOE (*p* = 0.12) (Figure 2H) was not significant.

Since the validated proteins were mostly plasma proteins, we wondered if these proteins are also expressed in endometrial cancer. To this aim, we inspected the Protein Atlas database to verify the expression of their genes in different tissues, and we found that the mRNAs codifying for all of these proteins are expressed in EC tissue. In addition, we performed WB analysis in protein lysates from three EC tissues to verify the expression of these proteins. As shown in Figure 3, proteins APOA1, CA1, HBB, HBD, and LPA were all found as single band, while SAA4 and PF4V1 were found as double band. The MW weight of these proteins in tissue corresponded to the apparent MW that was found through Western blotting of the serum exosome. This experiment confirmed that all the proteins were detectable in the endometrial tumor, and that proteoforms found in exosomes are probably secreted from tumor cells, suggesting that these could be bona fide EC biomarkers.

### 3.3. Statistical Results

The first step in developing a predictive model was to build a saturated multivariate logistic model with the outcome EC patients vs. controls. Results are reported in Table 2.

Following the step-down procedure, we obtained the model reported in Table 3. The latter yielded a pseudo R-squared = 0.277, an AUC = 83.5% (95% CI 74.1–92.9%), reaching a sensitivity of 82.86% with a specificity of 71.43% (Figure 4 and Figure 5). For the regression coefficients reported in Table 2, and the predicted probability cut points reported in Table 3 and Table 4, the following model identified cases and controls with the specified sensitivity and specificity:Predicted probability = 1/(1 + exp (−(−5.75242 + 0.7624538 × PF4V1 + 0.0467958 × APOE + 0.0738784 × HBD)))

Considering that this result was not fully satisfactory, we hypothesized that we could obtain better results by separating EC patients into Stage 1 and Advanced Stages (2, 3 and 4), and comparing these two groups separately with the control group.

### 3.4. Stage 1 EC Patients vs. Controls

We adopted the same approach as above. The results of the saturated multivariate logistic regression model with outcome Stage 1 EC patients vs. controls are reported in Table 5.

Following the step-down procedure, we obtained the model reported in Table 6, which yielded a pseudo R-squared = 0.717, an AUC = 98.0% (95% CI 95.0–100%), reaching a sensitivity of 100% with a specificity of 86.11% (Figure 6 and Figure 7). For the regression coefficients reported in Table 6 and the predicted probability cut points reported in Table 7, the following model identified cases and controls with the specified sensitivity and specificity:Predicted probability = 1/(1 + exp (−(−15.43866 + 1.689701 × PF4V1 + 0.1522074 × CA1 + 0.1571125 × HBD)))

### 3.5. Advanced Stage EC Patients vs. Controls

The results of the saturated multivariate logistic regression model with the outcome Advanced Stage EC patients vs. controls are reported in Table 8.

Following the step-down procedure, we obtained the model reported in Table 9, with only one independent variable (ApoE) remaining. The model yielded a pseudo R-squared = 0.201, and an AUC = 80.9% (95% CI 69.5–92.4%). It reached a sensitivity of 83.33% for a specificity of 68.57% (Figure 8 and Figure 9). For the regression coefficients reported in Table 9, and the predicted probability cut points reported in Table 10, the following model identified cases and controls with the specified sensitivity and specificity:Predicted probability = 1/(1 + exp (−(−2.929284 + 0.0540449 × ApoE)))(1)

### 3.6. Bioinformatic Analysis

For proteomic enrichment data analysis, we used gProfiler (Figure 10) tool classification that classified the proteins into groups according to their molecular function, biological processes, and protein class. Regarding biological processes, proteins were categorized into reverse cholesterol transport, cholesterol transport, cholesterol efflux, defense response, sterol transport, plasma lipoprotein particle remodeling, and protein–lipid complex remodeling. Regarding the molecular function, the proteins were classified in the following categories: cholesterol transfer activity, sterol transfer activity, phosphatidylcholine-sterol O-acyltransferase activator activity, lipoprotein particle receptor binding, antioxidant activity, sterol transporter activity, and lipid transfer activity. In addition, considering the cell compartment, the proteins were organized into blood microparticle, extracellular space, extracellular region, plasma lipoprotein particle, lipoprotein particle, protein–lipid complex, extracellular exosome, and extracellular vesicle. Looking at pathways, the Reactome tool analysis indicated that these proteins pertained to five pathways: plasma lipoprotein assembly, chylomicron assembly, chylomicron remodeling, plasma lipoprotein remodeling, hemostasis, plasma lipoprotein assembly, remodeling, clearance, and platelet degranulation.

IPA showed that these proteins are required in top networks corresponding to (Figure 11) (i) cell movement of leukocytes (LPA, PF4V1, VTN, APOD, APOA1, FGA, SERPINA3, APOE, and SERPING1); (ii) migration of cells (A2M, APOA1, APOB, APOD, CLU, FGA, LGALS3, LPA, PF4V1, PROS1, SERPINA3, SERPING1, THRB, and VTN); (iii) activation of phagocytes (APOA1, APOE, LGALS3, PROS1, SERPINF2, SERPING1, and VTN); and (iv) binding of antigen-presenting cells (A2M, APOA1, FGA, and APOE).

## 4. Discussion

Cancer biomarkers help to characterize tumor alterations, and are frequently used for the diagnosis and prognosis of the disease, and for determination of a personalized treatment [27]. Exosomes are microvesicles that, once released, play key roles in tumor growth and invasion. The proteomic characterization of patients’ exosomes is still challenging, but may represent a key step in the discovery of new potential biomarkers, particularly at an early stage [28]. LC-MS/MS has been successfully used to identify the proteomic profile of exosomes, and for biomarker identification as in prostate cancer [29], bladder cancer [30], and ovarian cancer serum [31]. In EC, thanks to proteomic approaches (such as two-dimensional electrophoresis, protein arrays, and mass spectrometry), hundreds of proteins have been reported as potential biomarkers in cancer tissue, blood, its derivatives, and other body fluids (17,18). Nevertheless, none of them have reached clinical stages, probably due to lack of tissue specificity, and also since most of them are proteins that are involved in broad processes including metabolic pathways, inflammatory responses, cell adhesion, and in hormones. To find specific non-invasive biomarkers, there is growing interest in exploring exosome-enriched proteins.

Interestingly, Song et al. recently found that plasma exosomes from EC patients are enriched in LGALS3BP, a protein that also promotes endometrial cancer progression.

In this study, we used a larger cohort of EC patients to identify novel exosomal biomarkers, and investigated the exosome proteome from albumin-depleted serum, using LFQ based proteomics followed by Western blotting analysis validation.

Starting with the proteomic profile, we obtained 440 proteins that were further selected based on the strongest differences in abundances; then, we applied a multivariate logistic regression analysis for all EC stages (1, 2, 3, and 4). By this analysis, we found that PF4V1, APOE, and HBD allowed us to separate cases from controls with an AUC = 83.5% in 36 EC patients as compared to 36 healthy individuals.

Considering that this result was not fully satisfactory, and attempting to find early EC biomarkers, we decided to separate patients into stage 1 (often asymptomatic patients) and advanced stages (2, 3, and 4).

A multivariate logistic regression model for stage 1 (18 patients) based on PF4V1, CA1, and HBD allowed us to separate cases from controls with an AUC = 98.0%. The last multivariate logistic regression model was performed that compared patients with an advanced stage EC (2, 3, or 4) (18 patients). Based on ApoE expression, it allowed us to separate cases from controls with an AUC = 80.9%. The best and only fully satisfactory model was that of the analysis that considered stage 1 patients. It is noteworthy that proteins that discriminate stage 1 ECs well do not satisfactorily discriminate all stages of EC. The low performance of these proteins in the more advanced stages of the disease is interesting, and requires further investigation. This study shows that stage 1 discriminating proteins do not work well for more advanced stages of EC.

Reactome analysis revealed that EC exosome proteins are involved in dysregulation of plasma lipoprotein assembly and remodeling, hemostasis, and platelet degranulation pathways that may be involved in cancer development.

CA1 is a member of the carbonic anhydrase (CA) family, and an overexpression in osteosarcoma cells leads to calcification with ascorbic acid [32]. Wang et al., in a proteomic study, identified this protein in stage I non-small cell lung cancer, and validated its overexpression via Western blotting; this represents a promising early biomarker for non-small cell lung cancer.

HBB is involved in oxygen transport from the lung to several peripheral tissues [33]. Expression of HBB in lung cancer cells and breast cancer cells is associated with ROS cytotoxicity suppression, leading to cancer cell survival and spread [34].

PF4V1 suppresses chemokine angiogenesis by blocking the protein bFGF, and is closely associated with the growth and metastasis of various cancers [35]. It is known that PF4V1 in prostate cancer leads to suppression of proliferation and invasion, and serves as a potential prognostic biomarker [36].

APOE is a protein associated with lipid particles, that mainly functions in lipoprotein-mediated lipid transport between organs via the plasma and interstitial fluid [37]. Studies on several tumors, including glioblastoma [38], EC [39], lung cancer [40], and prostate cancer [41], showed that when APOE is overexpressed, the disease is more aggressive, and the prognosis is poor [42].

Of note, although these proteins are commonly found in plasma upon liver secretion, in this study we found that they are detectable in endometrial tumor tissue; thus, they may represent bona fide EC biomarkers when secreted into exosomes by tumor cells.

Interestingly, through Western blotting we noticed that CA1, PF4V1, and APOE displayed higher molecular weights than the canonical ones. This may rely on the glycosylation of proteins in exosomes, as previously described by the literature [43]. Whether glycosylation may affect protein functions or exosome distribution is still unknown, and is of great interest.

Importantly, in this study we provided a proteomic profile of EC serum exosomes that suggests new promising non-invasive biomarkers, although we are aware of some limitations. These proteins have not yielded satisfactory results for all stages together, or for the more advanced stages of EC. However, in the discovery phase, we identified proteins that discriminated well between controls and stage 1 patients. This suggests that for advanced stages, separate studies are needed, as biomarkers that work well for stage 1 are not necessarily efficient for advanced stages or all stages together. Moreover, we are aware that broader studies are needed to validate the role of the identified proteins in larger cohorts of patients. Another weakness of our study is that patients of the discovery phase were also included in the validation phase, which, however, was performed in a larger cohort that included 24 patients that did not overlap with the previous ones. Lastly, since levels of some proteins may change with age, we think that validating these results with age-matched cases and controls should be the next step to ascertain the potential of these proteins as biomarkers.

## 5. Conclusions

In our opinion, our proteomic data may expand the knowledge of the protein composition of EC serum exosomes, and may contribute to the discovery of new EC biomarkers. Optimally, the proteins that we validated were able to discriminate stage 1 EC patients from controls, but failed to satisfactorily identify EC patients at more advanced stages. Other studies with larger cohorts of patients are necessary to develop new algorithms that are able to discriminate patients at more advanced stages, while matching cases and controls by age. LC-MS/MS represents a powerful technology to explore the serum exosome, allowing for the identification of candidate biomarkers in EC.

## Figures and Tables

**Figure 1 cancers-14-06262-f001:**
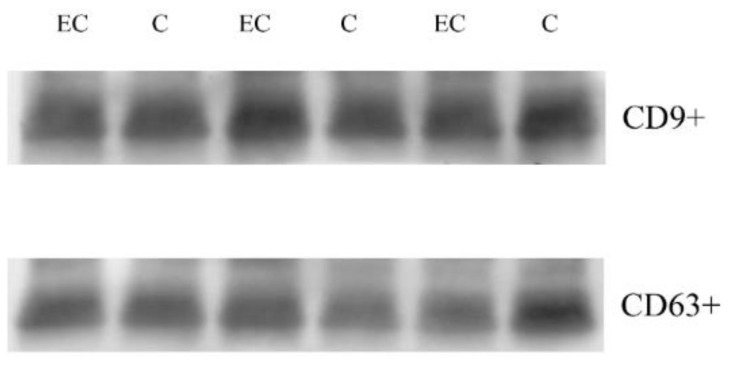
CD 63 and CD9 common exosome markers. Western blotting in exosome of three C (control) and three EC (endometrial cancers).

**Figure 2 cancers-14-06262-f002:**
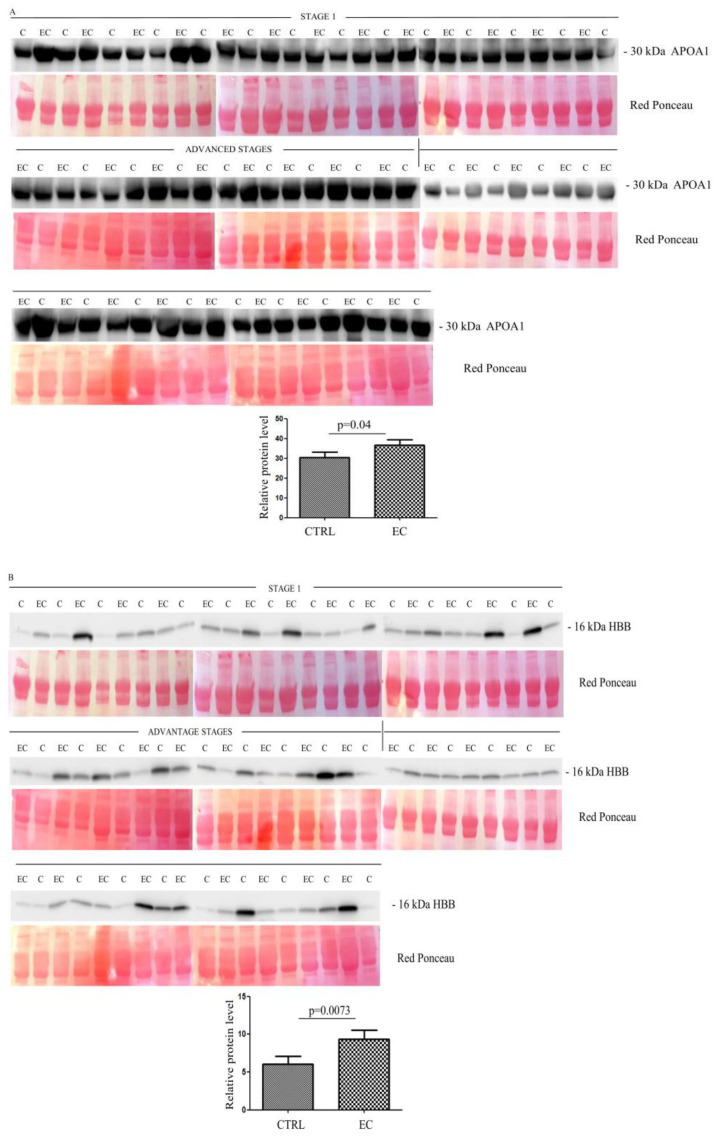
(**A**–**H**) Western blotting analyses of exosomes for eight proteins APOA1, HBB, CA1, HBD, LPA, SAA4, PF4V1, and APOE in controls (C) and endometrial cancer (EC) patients. The intensity of immunostained bands was normalized against the total protein intensities measured from the same blot stained with Red Ponceau. The graph shows the relative abundance of the proteins in control and endometrial cancer exosomes. Results are shown as a histogram (*p* < 0.05), with each bar representing mean ± standard deviation.

**Figure 3 cancers-14-06262-f003:**
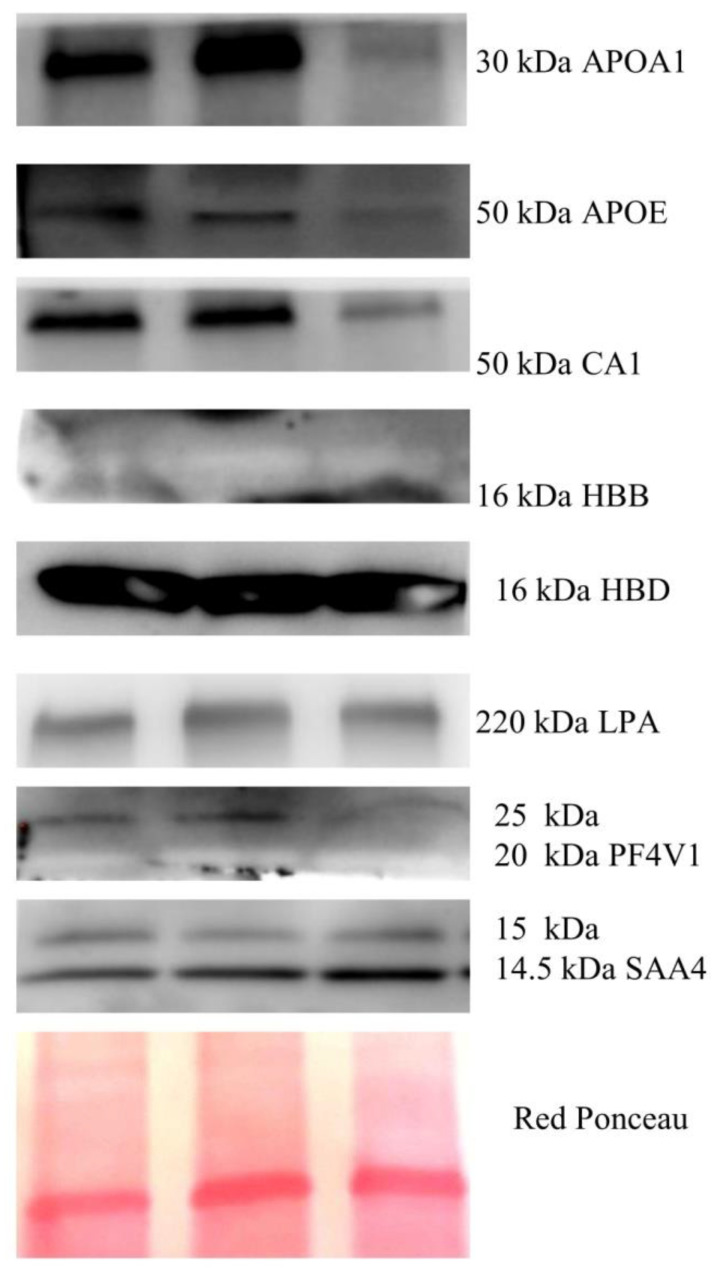
Western blotting of proteins APOA1, APOE, CA1, HBB, HBD, LPA, PF4V1, and SAA4 present in EC.

**Figure 4 cancers-14-06262-f004:**
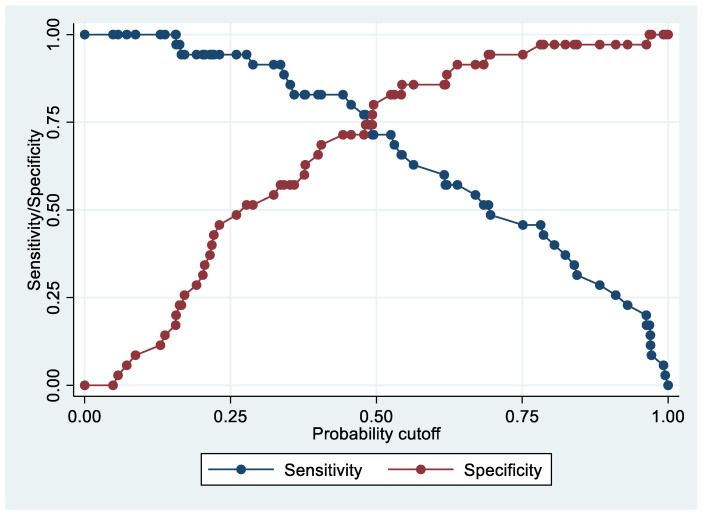
Sensitivity and specificity plot of the final model with the outcome EC patients vs. controls.

**Figure 5 cancers-14-06262-f005:**
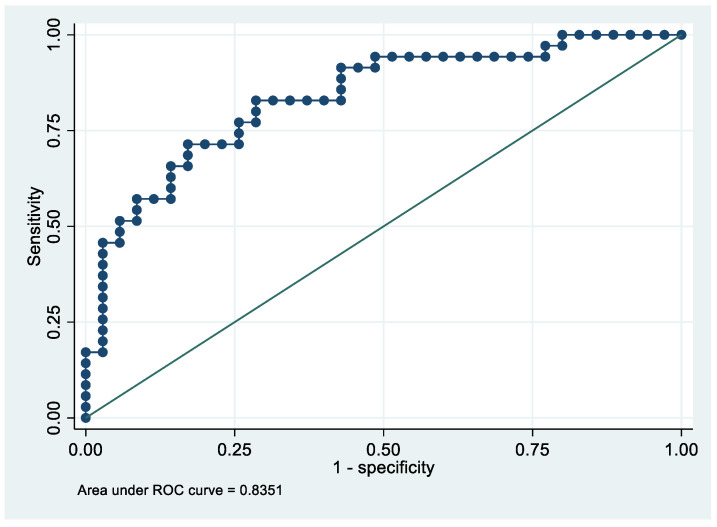
Receiver operating characteristics curve of the final model with the outcome EC patients vs. controls.

**Figure 6 cancers-14-06262-f006:**
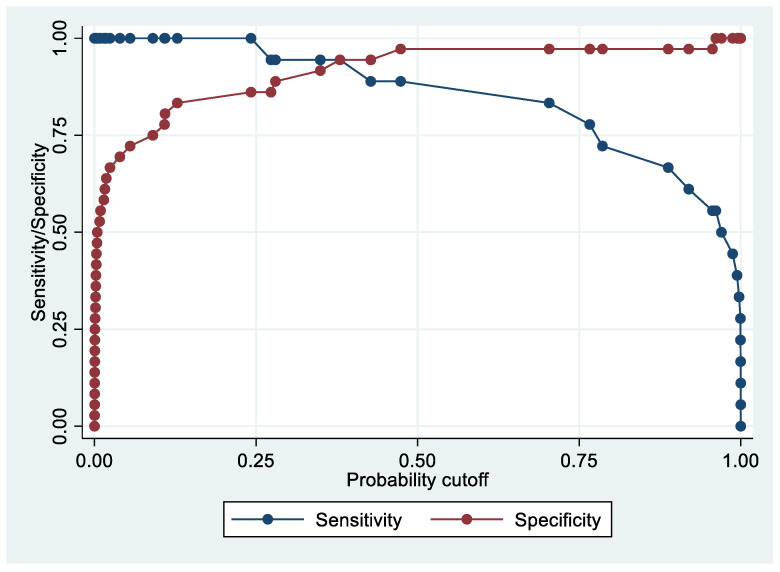
Sensitivity and specificity plot of the final model with the outcome Stage 1 EC patients vs. controls.

**Figure 7 cancers-14-06262-f007:**
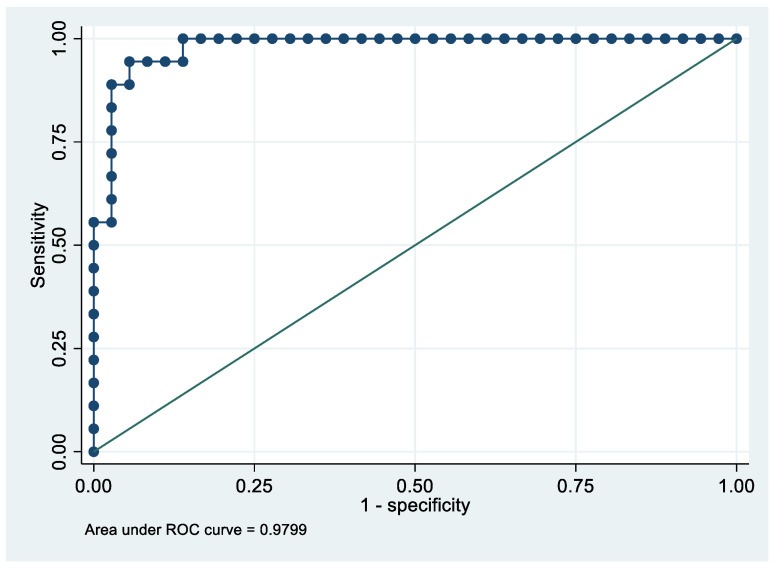
Receiver operating characteristics curve of the final model with the outcome Stage 1 EC patients vs. controls.

**Figure 8 cancers-14-06262-f008:**
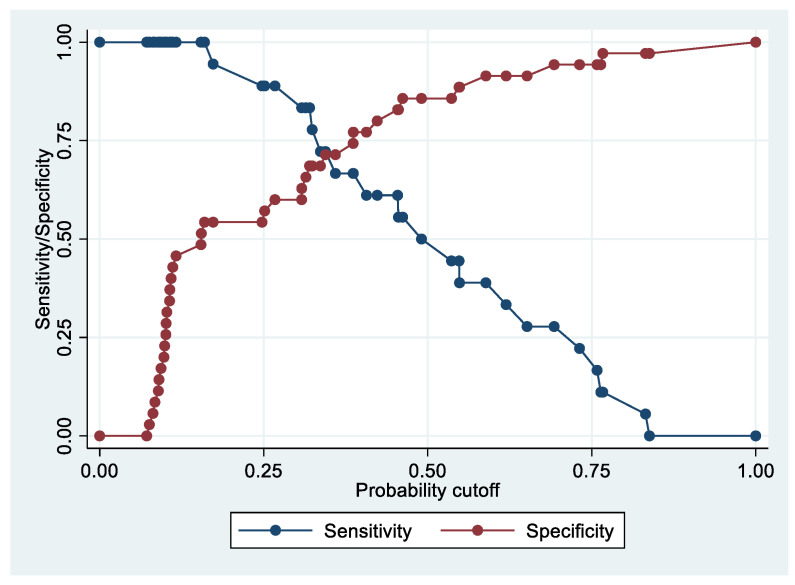
Sensitivity and specificity plot of the final model with the outcome Advanced Stage EC patients vs. controls.

**Figure 9 cancers-14-06262-f009:**
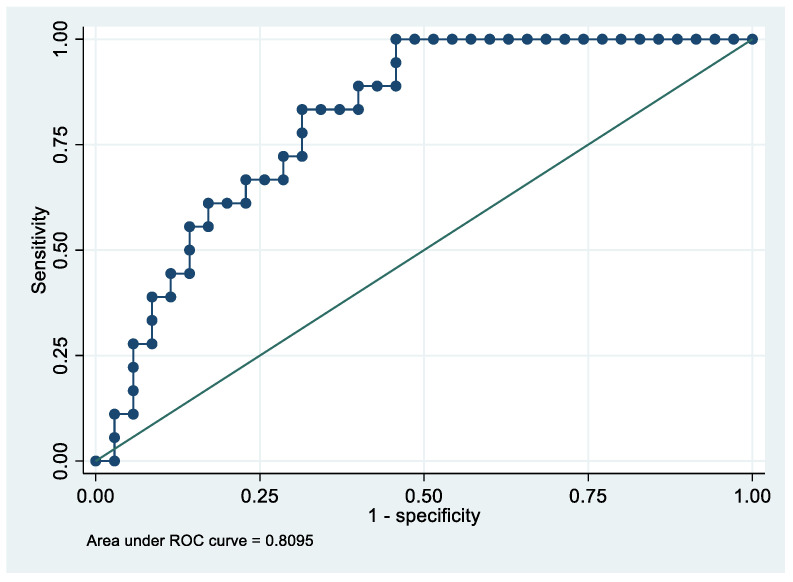
Receiver operating characteristics curve of the final model with the outcome Advanced Stage EC patients vs. controls.

**Figure 10 cancers-14-06262-f010:**
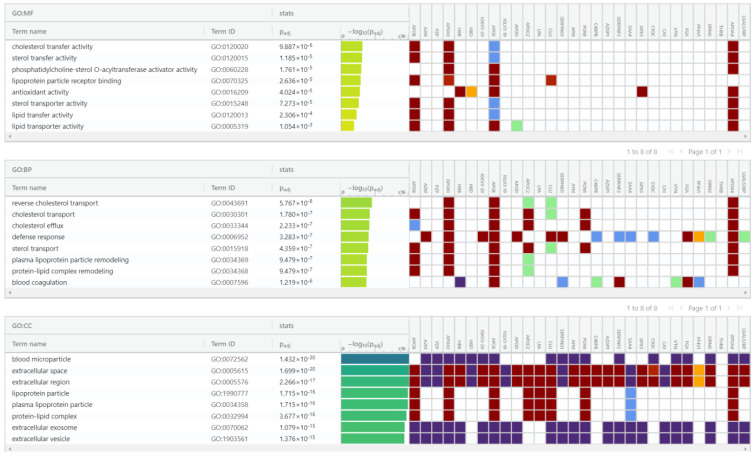
gProfiler classification of proteins in the EC serum exosomes according to their molecular function, biological processes, and cellular component.

**Figure 11 cancers-14-06262-f011:**
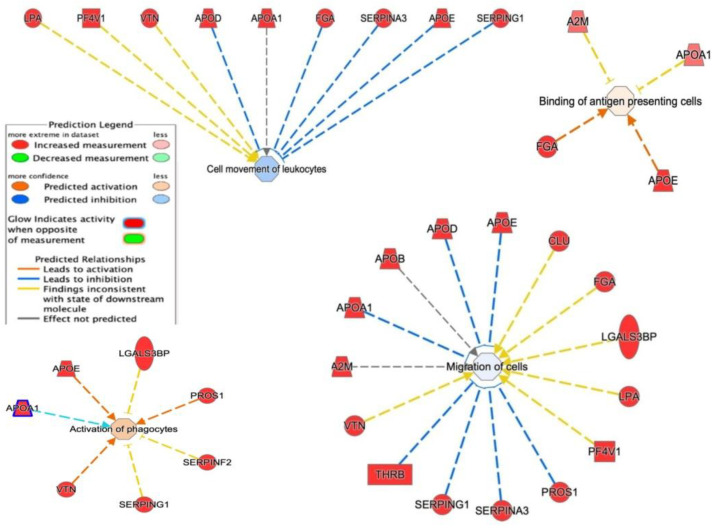
Network build-up from one of the most significant bio-functions: (1) activation of phagocytosis; (2) binding of antigen-presenting cells; (3) cell movement of leukocytes; (4) migration of cells.

**Table 1 cancers-14-06262-t001:** Different abundance proteins identified by mass spectrometry in EC compared to the control serum exosome.

Accession Number	Protein Description	Gene Symbol	Fold Change	*p*-Value
P00915	Carbonic anhydrase 1	CA1	9.0	0.0086
P02042	Hemoglobin subunit delta	HBD	6.7	0.0077
P68871	Hemoglobin subunit beta	HBB	5.88	0.018
P08519	Apolipoprotein(a)	LPA	4.65	0.046
P10720	Platelet factor 4 variant	PF4V1	3.69	0.001
P35542	Serum amyloid A-4 protein	SAA4	3.66	0.002
P02647	Apolipoprotein A-I	APOA1	3.6	0.04
P02649	Apolipoprotein E	APOE	3.5	0.0086
P69905	Hemoglobin subunit alpha	HBA1	3.45	0.0036
P02747	Complement C1q subcomponent subunit C	C1QC	2.62	0.035
P20851	C4b-binding protein beta chain	C4BPB	2.5	0.022
P02655	Apolipoprotein C-II	APOC2	2.5	0.016
P04004	Vitronectin	VTN	2.44	0.022
P02671	Fibrinogen alpha chain	FGA	2.33	0.012
P19652	Alpha-1-acid glycoprotein 2	ORM2	2.07	0.04
P04114	Apolipoprotein B-100	APOB	2.0	0.046
P07225	Vitamin K-dependent protein S	PROS	2.0	0.026
P01011	Alpha-1-antichymotrypsin	AACT	1.95	0.046
H0YAS8	Clusterin	CLU	1.85	0.04
P05090	Apolipoprotein D	APOD	1.8	0.03
P00734	Prothrombin	THRB	1.77	0.019
P08697	Alpha-2-antiplasmin	SERPINF2	1.76	0.026
P22352	Glutathione peroxidase 3	GPX3	1.75	0.03
P25311	Zinc-alpha-2-glycoprotein	AZGP1	1.75	0.046
P05155	Plasma protease C1 inhibitor	SERPING1	1.71	0.035
P27169	Serum paraoxonase/arylesterase 1	PON1	1.69	0.035
P01023	Alpha-2-macroglobulin	A2MG	1.67	0.03
Q08380	Galectin-3-binding protein	LGALS3BP	1.66	0.035
P43652	Afamin	AFM	1.57	0.019
P06727	Apolipoprotein A-IV	APOA4	1.51	0.0086
P20742	Pregnancy zone protein	PZP	1.5	0.046
P01714	Immunoglobulin lambda variable 3–19	IGLV3–19	0.66	0.04
P01619	Immunoglobulin kappa variable 3–20	IGKV3–20	0.6	0.01

**Table 2 cancers-14-06262-t002:** Saturated multivariate logistic regression model including all selected exosomes (ApoA, HBB, CA1, LPA, SAA4, PF4V1, ApoE, and HBD), with the outcome EC patients vs. controls.

Genes	OR (95% CI)	*p*-Value
PF4V1	1.800 (1.071–3.026)	0.027
APOE	1.040 (1.002–1.080)	0.040
SAA4	1.030 (0.993–1.070)	0.117
HBD	1.075 (0.979–1.181)	0.127
CA1	1.064 (0.981–1.153)	0.134
HBB	1.085 (0.973–1.210)	0.144
APOA	1.015 (0.968–1.064)	0.538
LPA	1.057 (0.838–1.335)	0.638

Note: OR = odds ratio; CI = confidence interval.

**Table 3 cancers-14-06262-t003:** Final multivariate logistic regression model including only exosomes significantly associated with the outcome EC patients vs. controls.

Genes	*p*-Value	OR (95% CI)	Coefficient (95% CI)
PF4V1	0.001	2.144 (1.366–3.363)	0.7624538 (0.3120302–1.212877)
APOE	0.006	1.048 (1.013–1.084)	0.0467958 (0.0132013–0.0803904)
HBD	0.034	1.077 (1.006–1.153)	0.0738784 (0.0056874–0.1420693)
Constant	0.000	0.003 (0.000–0.065)	−5.75242 (−8.771762–−2.733079)

Note: OR = odds ratio; CI = confidence interval; Coefficient = logistic regression model coefficient.

**Table 4 cancers-14-06262-t004:** Final EC patients vs. controls model: sensitivity/specificity plot for the resulting multivariate logistic regression model for sensitivity levels higher than specificity levels.

Predicted Probability Cut Point	Sensitivity	Specificity
≥0.1564960	100.00%	20.00%
≥0.1626552	97.14%	22.86%
≥0.2775130	94.29%	51.43%
≥0.3356628	91.43%	57.14%
≥0.4427229	82.86%	71.43%

**Table 5 cancers-14-06262-t005:** Saturated multivariate logistic regression model including all selected exosomes (APOA, HBB, CA1, LPA, SAA4, PF4V1, APOE, and HBD), with the outcome Stage 1 EC patients vs. controls.

Genes	OR (95% CI)	*p*-Value
HBB	1.155 (0.975–1.369)	0.095
PF4V1	2.234 (0.864–5.777)	0.097
HBD	1.030 (0.973–1.090)	0.307
CA1	1.054 (0.952–1.168)	0.308
APOA	1.027 (0.941–1.120)	0.555
LPA	1.173 (0.656–2.099)	0.591
SAA4	1.026 (0.933–1.127)	0.599
ApoE	0.999 (0.925–1.079)	0.978

Note: OR = odds ratio; CI = confidence interval.

**Table 6 cancers-14-06262-t006:** Final multivariate logistic regression model including only exosomes significantly associated with the outcome Stage 1 EC patients vs. controls.

Genes	*p*-Value	OR (95% CI)	Coefficient (95% CI)
PF4V1	0.010	5.418 (1.486–19.751)	1.689701 (0.3961953–2.983206)
CA1	0.013	1.164 (1.032–1.314)	0.1522074 (0.0316167–0.2727981)
HBD	0.047	1.070 (1.002–1.366)	0.1571125 (0.0021436–0.3120813)
Constant	0.003	1.97 × 10^−7^ (6.40 × 10^−12^–0.006)	−15.43866 (−25.77476–−5.102551)

Note: OR = odds ratio; CI = confidence interval; Coefficient = logistic regression model coefficient.

**Table 7 cancers-14-06262-t007:** Final Stage 1 EC patients vs. controls model: sensitivity/specificity plot for the resulting multivariate logistic regression model for sensitivity levels higher than specificity levels.

Predicted Probability Cut Point	Sensitivity	Specificity
≥0.2421473	100.00%	86.11%
≥0.3798218	94.44%	94.44%

**Table 8 cancers-14-06262-t008:** Saturated multivariate logistic regression model including all selected exosomes (ApoA, HBB, CA1, Lp(a), SAA4, PF4V1, ApoE, and HBD), with the outcome Advanced Stage EC patients vs. controls.

Genes	OR (95% CI)	*p*-Value
APOE	1.048 (1.007–1.091)	0.021
SAA4	1.024 (0.987–1.063)	0.200
HBD	1.068 (0.965–1.182)	0.202
HBB	1.058 (0.950–1.177)	0.304
PF4V1	1.175 (0.588–2.349)	0.648
LPA	1.048 (0.821–1.338)	0.705
CA1	1.017 (0.913–1.133)	0.760
ApoA	0.993 (0.931–1.059)	0.840

Note: OR = odds ratio; CI = confidence interval.

**Table 9 cancers-14-06262-t009:** Final multivariate logistic regression model including only exosomes significantly associated with the outcome Advanced Stage EC patients vs. controls.

Genes	*p*-Value	OR (95% CI)	Coefficient (95% CI)
APOE	0.001	1.055 (1.021–1.091)	0.0540449 (0.0208706–0.0872192)
Constant	0.000	0.053 (0.011–0.269)	−2.929284 (−4.546729–−1.311839)

Note: OR = odds ratio; CI = confidence interval; Coefficient = logistic regression model coefficient.

**Table 10 cancers-14-06262-t010:** Final Advanced Stage EC patients vs. controls model: sensitivity/specificity plot for the resulting multivariate logistic regression model for sensitivity levels higher than specificity levels.

Predicted Probability Cut Point	Sensitivity	Specificity
≥0.159793	100.00%	54.29%
≥2671622	88.89%	60.00%
≥3199555	83.33%	68.57%
≥0.344161	72.22%	71.43

## Data Availability

The data presented in this study are available on request from the corresponding author. The data are not publicly available due to ethical reasons.

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
