# Peer review of "A Label-Free Proteomic Approach for the Identification of Biomarkers in the Exosome of Endometrial Cancer Serum"

_cancers, 2022, doi:10.3390/cancers14246262_

Round 1
Reviewer 1 Report
An overall interesting, well performed and described study of interest to specialized readers. Results are clearly presented. There are a couple of issues that needs to be addressed before publication.
The supplementary table 1 only cover 50% of the patients. A table with complete patient characteristics and histology is absolutely needed.
The authors need to describe which patients are in the discovery phase and which patients are in the validation phase. Are patients from the discovery phase used in the validation phase? This should be clearly described. And if this is the case the validation is hampered by the results from the discovery and leads to unprecedented, good results compared to a totally external cohort validation. This should be thoroughly discussed as a weakness of the study.
The patients included in the sub analysis of stage 1-2 and 3-4 also need to be specified. How many? What histology, grade, age and so on.
An external validation is needed to confirm the data from this study, and this should be emphasized in the discussion and conclusion.
Author Response
Reviewer 1
The supplementary table 1 only cover 50% of the patients. A table with complete patient characteristics and histology is absolutely needed.
Our reply: We completed table 1 following the Reviewer’s suggestions.
The authors need to describe which patients are in the discovery phase and which patients are in the validation phase. Are patients from the discovery phase used in the validation phase? This should be clearly described. And if this is the case the validation is hampered by the results from the discovery and leads to unprecedented, good results compared to a totally external cohort validation. This should be thoroughly discussed as a weakness of the study. The patients included in the sub analysis of stage 1-2 and 3-4 also need to be specified. How many? What histology, grade, age and so on.
Our reply: The patients used in the discovery phase are 12, all carrying EC at stage 1 (patient numbers: 152, 157, 190, 135, 165, 166, 167, 168, 170, 181, 183, 140). To confirm MS data, all patients from the discovery phase were then also used in the validation phase. In addition, we enlarged our analysis with a cohort of 24 patients not overlapping with the previous ones (6 stage 1 and 18 advanced stages). In the table 1 we added all the information regarding the patients. These aspects were now dealt with in the discussion and conclusion sections of the manuscript.
An external validation is needed to confirm the data from this study, and this should be emphasized in the discussion and conclusion.
Our reply: The external validation was made on 24 novel EC patients (6 with EC at stage 1 and 18 at advanced stages).

Reviewer 2 Report
The manuscript entitled by “A Label-Free Proteomic Approach for the Identification of Biomarkers in the Exosome of Endometrial Cancer Serum” from Sommella et al. presents a label-free proteomics analysis of serum exosome from EC patients and healthy women in together with step-wise statistical analysis to identify a protein panel for early diagnosis of EC. This work is important for the clinical EC diagnosis and may be interested to our readers, however, this manuscript is not well-written and several issues should be further revised or addressed. Following are comments and suggestions:
1. In general, the manuscript is not well-written and examined before submission. Many typos and typing errors could be found throughout the manuscript, such as missing blank in Table1, Fig.1 (line 184), Fig.2 (line 207-209), etc. Typos like “and” should be “an” in abstract (line 30); Resolution should be 120,000 and 15,000 instead of 120.000 and 15.000 (line 129); 3 Da (line 130), “ad” should be “and” (line 166). The blank in “0.1 %” (line 125), “4 °C” (line 144), CD 163 (line 186) should be deleted. Inconsistent wording such as “μl” and “μL” (line 107, 110); “table 2” and “Table 2” (line 195); LC-MS/MS and LC/MS-MS (line 338), Lp(a) and LPA, PF4V1 and pf4v1 in tables and table legends are frequently found. There are still many other errors in the manuscript which is not listed here, please examine the entire manuscript carefully to correct these errors. More importantly, the numbering of Tables is not correct; Table 7, 8, and 9 are missing, instead, the author used Table 10, 11, 12.
2. In the introduction, the authors should introduce previous studies which reported the discovery of biomarkers using proteome approaches for EC detection and provide discussions toward their results and the previous findings.
3. In this study, the authors used label-free quantitation approach to study the exome proteome, but the details about the quantitation settings or parameters are not included in the method. Besides, only single unique peptide is used for protein identification might be okay, but it would be more reliable to adopt at least two unique peptides for quantitation.
4. It’s good that the authors validated the expression levels of candidate proteins in another cohort of 18 EC and 18 control samples. However, the presentation and analysis of validation data in Figure 2 is not convincing. The authors mentioned they used the total intensity in ponceau image as internal control to quantify the target protein expressions between EC and control samples, but several ponceau images have obvious interferences/background which are not good internal standard or may cause incorrect normalization. The authors should offer the results with actin or tubulin as internal control as well as provide the ponceau images of entire membrane for each protein. They should clarify whether these results were obtained from the same or different membranes. In addition, several ponseau images look the same; for example, the ponceau images in APOA1 is the same as those in HBB, CA1, SSA4, and PF4V1. They mentioned these western blotting results with 2A, 2B, 2C… etc, but there is no such labeling in Figure 2. It’s also difficult to know what are the results from Stage-I or advanced ECs.
5. The resolution of the Figure 10 is quite poor. Bigger font should be used to make all the protein and network names clearer. The authors used I, ii, iii to describe these networks in the content, but in the figure legend, 1, 2, 3 were used. It’s hard to follow their story. Besides, the grammar in description of Figure 10 in line 322 -327 should be revised.
6. The manuscript may require English editing to enhance the better reading and understanding of their findings.
Other minor suggestions:
1. Line 30, use for “an” early stage
2. Line 63, add “.” after woman
3. Line 67: add space between “(D&C) to”
4. The affiliation should be revised according to the authorship.
Author Response
- In general, the manuscript is not well-written and examined before submission. Many typos and typing errors could be found throughout the manuscript, such as missing blank in Table1, Fig.1 (line 184), Fig.2 (line 207-209), etc. Typos like “and” should be “an” in abstract (line 30); Resolution should be 120,000 and 15,000 instead of 120.000 and 15.000 (line 129); 3 Da (line 130), “ad” should be “and” (line 166). The blank in “0.1 %” (line 125), “4 °C” (line 144), CD 163 (line 186) should be deleted. Inconsistent wording such as “μl” and “μL” (line 107, 110); “table 2” and “Table 2” (line 195); LC-MS/MS and LC/MS-MS (line 338), Lp(a) and LPA, PF4V1 and pf4v1 in tables and table legends are frequently found. There are still many other errors in the manuscript which is not listed here, please examine the entire manuscript carefully to correct these errors. More importantly, the numbering of Tables is not correct; Table 7, 8, and 9 are missing, instead, the author used Table 10, 11, 12.
Our reply: We fixed all the errors in the manuscript.
- In the introduction, the authors should introduce previous studies which reported the discovery of biomarkers using proteome approaches for EC detection and provide discussions toward their results and the previous findings.
Our reply: We expanded both the introduction and the discussions, following the Reviewer’s suggestions.
- In this study, the authors used label-free quantitation approach to study the exome proteome, but the details about the quantitation settings or parameters are not included in the method. Besides, only single unique peptide is used for protein identification might be okay, but it would be more reliable to adopt at least two unique peptides for quantitation.
Our reply: The label free quantification has been performed with a Mascot Distiller software and it is based on a Replicate protocol. This workflow is based on the relative intensities of high resolution extracted ion chromatograms (XICs) for precursor ions in multiple data sets and aligned using mass and elution time. Relative quantitation is based on protein ratio calculation which uses the median of the assigned peptide ratios. Minimum precursor charge is set to 2 and the minimum peptides number is set to 2. Replicate protocol is used to measure the relative abundance of a protein from sample to sample. This description has now been added to the manuscript.
- It’s good that the authors validated the expression levels of candidate proteins in another cohort of 18 EC and 18 control samples. However, the presentation and analysis of validation data in Figure 2 is not convincing. The authors mentioned they used the total intensity in ponceau image as internal control to quantify the target protein expressions between EC and control samples, but several ponceau images have obvious interferences/background which are not good internal standard or may cause incorrect normalization. The authors should offer the results with actin or tubulin as internal control as well as provide the ponceau images of entire membrane for each protein. They should clarify whether these results were obtained from the same or different membranes. In addition, several ponseau images look the same; for example, the ponceau images in APOA1 is the same as those in HBB, CA1, SSA4, and PF4V1. They mentioned these western blotting results with 2A, 2B, 2C… etc, but there is no such labeling in Figure 2. It’s also difficult to know what are the results from Stage-I or advanced ECs.
Our reply: To normalize the results of the WB analysis, we decided to determine the total protein content of each sample by Red Ponceau, since increasing data in the literature reported that total protein normalization outperform housekeeping protein immunodetection as loading controls (B. Rivero-Gutiérrez, A. Anzola, O. Martínez-Augustin, F. Sánchez de Medina. Stain-free detection as loading control alternative to Ponceau and housekeeping protein immunodetection in Western blotting, Analytical Biochemistry, Volume 467, 2014,) (Sander H, Wallace S, Plouse R, Tiwari S, Gomes AV. Ponceau S waste: Ponceau S staining for total protein normalization. Anal Biochem. 2019 Jun 15;575:44-53.). Moreover, in EC most of the proteins commonly used as housekeeping are indeed affected and thus not adequate to be used as a control for normalization. For example, according to our previous published works, GAPDH is up-regulated in EC (Ura B, Monasta L, Arrigoni G, Franchin C, Radillo O, Peterlunger I, Ricci G, Scrimin F: A proteomic approach for the identification of biomarkers in endometrial cancer uterine aspirate. Oncotarget. 2017 Dec 12; 8: 109536–109545), while ACTB is down-regulated in the serum of EC patients (Ura B, Biffi S, Monasta L, Arrigoni G, Battisti I, Di Lorenzo G, Federico Romano F, Aloisio M, Celsi F, Addobbati R, Valle F, RampazzoE , Brucale M, ARidolfi A, Licastro D, Ricci G: Two Dimensional-Difference in Gel Electrophoresis (2D-DIGE) Proteomic Approach for the Identification of Biomarkers in Endometrial Cancer Serum. Cancers 2021, 13, 3639). Similarly, the tubulin has also been found up-regulated in EC (Charles Dumontet; Mary Ann Jordan; Francis F.Y. Lee. Ixabepilone: targeting βIII-tubulin expression in taxane-resistant malignancies. Mol Cancer Ther (2009) 8 (1): 17–25). We thus decided to apply a total protein content normalization method, because we could not establish which proteins should be considered as good housekeeping in our samples.
The ponceau stainings shown were obtained from the same membranes where the immunoblot has been performed. As required by the Reviewer, we provided the ponceau images of the entire membrane for each protein, and we fixed the western blotting figures.
- The resolution of the Figure 10 is quite poor. Bigger font should be used to make all the protein and network names clearer. The authors used I, ii, iii to describe these networks in the content, but in the figure legend, 1, 2, 3 were used. It’s hard to follow their story. Besides, the grammar in description of Figure 10 in line 322 -327 should be revised.
Our reply: We fixed Figure 10 according to the Reviewer’s indications. We reviewed the grammar of the manuscript.
- The manuscript may require English editing to enhance the better reading and understanding of their findings.
Our reply: English editing was performed on the entire manuscript.
Other minor suggestions:
- Line 30, use for “an” early stage
- Line 63, add “.” after woman
- Line 67: add space between “(D&C) to”
- The affiliation should be revised according to the authorship.
Our reply: We fixed the errors in the manuscript.
Reviewer 3 Report
The authors have investigated the serum derived exsosomal proteins as a source of biomarkers from EC patients. The approach is attractive however, the final significant candidates APOA1, HBB, CA1, HBD, LPA, SAA4, PF4V1 and APOE as well as others listed in Table 1 are mostly plasma high abundant proteins secreted from liver possibly because of the inflammatory response not limited to cancer. The authors should compare their serum exosomal proteins with tumor driven gene expression and/or protein expression data since the potential benefit of exosome purification is the enrichment of tumor signature more efficiently than looking at the total serum circulating proteins which has huge dynamic concentration range.
Author Response
The authors have investigated the serum derived exsosomal proteins as a source of biomarkers from EC patients. The approach is attractive however, the final significant candidates APOA1, HBB, CA1, HBD, LPA, SAA4, PF4V1 and APOE as well as others listed in Table 1 are mostly plasma high abundant proteins secreted from liver possibly because of the inflammatory response not limited to cancer. The authors should compare their serum exosomal proteins with tumor driven gene expression and/or protein expression data since the potential benefit of exosome purification is the enrichment of tumor signature more efficiently than looking at the total serum circulating proteins which has huge dynamic concentration range.
Our reply: We thank the Reviewer for this observation. We agree with the Reviewer that the APOA1, HBB, CA1, HBD, LPA, SAA4, PF4V1 and APOE proteins are mostly plasmatic proteins. However, we inspected the Protein Atlas database to verify the expression of their genes in different tissues. We found that the mRNAs codifying for all these proteins are expressed in the EC tissue. Concordantly, we conduct western blotting in protein lysates from 3 EC tissue to verify the expression of these proteins. We confirmed that all the proteins were detectable in endometrial tumor. This observation has now been added in the text.
Round 2
Reviewer 1 Report
My comments are addressed adequately.
Reviewer 2 Report
The authors improved significantly the content and the labeling of figures and Tables. I have no further question for the manuscript. However, some minor test errors are still detected. Please go through the manuscript carefully and revise it before publication.
Reviewer 3 Report
The final proteins are the typical plasma abundant circulating proteins which you may easily detect without exosome purification. Nome of them are EC tumor exclusive proteins and I suspect they are contaminated or sticking to the exctracellular vesicles. I would suggest to add EC cell lines, cell lysate and the exosomes confirmation which would be pure system to support. Otherwise the authors may want to select non-plasma abundant proteins as marker candidates in your list.